# Point-of-Care Testing to Differentiate Various Acid–Base Disorders in Chronic Kidney Disease

**DOI:** 10.3390/diagnostics13213367

**Published:** 2023-11-02

**Authors:** Tomasz Gołębiowski, Sławomir Zmonarski, Wiktoria Rożek, Mateusz Powązka, Patryk Jerzak, Maciej Gołębiowski, Mariusz Kusztal, Piotr Olczyk, Jakub Stojanowski, Krzysztof Letachowicz, Mirosław Banasik, Andrzej Konieczny, Magdalena Krajewska

**Affiliations:** Department of Nephrology and Transplantation Medicine, Wroclaw Medical University, Borowska 213, 50-556 Wroclaw, Poland; slawomir.zmonarski@umw.edu.pl (S.Z.); wiktoria.rozek@onet.pl (W.R.); lek.mateuszpowazka@gmail.com (M.P.); patryk_jerzak@o2.pl (P.J.); maciej.golebiowski@student.umw.edu.pl (M.G.); mariusz.kusztal@umw.edu.pl (M.K.); p.olczyk@umw.edu.pl (P.O.); jakub.stojanowski@student.umw.edu.pl (J.S.); krzysztof.letachowicz@umw.edu.pl (K.L.); miroslaw.banasik@umw.edu.pl (M.B.); andrzej.konieczny@umw.edu.pl (A.K.); magdalena.krajewska@umw.edu.pl (M.K.)

**Keywords:** metabolic acidosis, CKD, bicarbonate, chloride

## Abstract

Background: Normal-anion-gap metabolic acidosis (AGMA) and high-anion-gap metabolic acidosis (HAGMA) are two forms of metabolic acidosis, which is a common complication in patients with chronic kidney disease (CKD). The aim of this study is to identify the prevalence of various acid–base disorders in patients with advanced CKD using point-of-care testing (POCT) and to determine the relationship between POCT parameters. Methods: In a group of 116 patients with CKD in stages G4 and G5, with a mean age of 62.5 ± 17 years, a sample of arterial blood was taken during the arteriovenous fistula procedure for POCT, which enables an assessment of the most important parameters of acid–base balance, including: pH, base excess (BE), bicarbonate (HCO_3_^−^), chloride(Cl^−^), anion gap (AG), creatinine and urea concentration. Based on this test, patients were categorized according to the type of acidosis-base disorder. Results: Decompensate acidosis with a pH < 7.35 was found in 68 (59%) patients. Metabolic acidosis (MA), defined as the concentration of HCO_3_^−^ ≤ 22 mmol/L, was found in 92 (79%) patients. In this group, significantly lower pH, BE, HCO_3_^−^ and Cl^−^ concentrations were found. In group of MA patients, AGMA and HAGMA was observed in 48 (52%) and 44 (48%) of patients, respectively. The mean creatinine was significantly lower in the AGMA group compared to the HAGMA group (4.91 vs. 5.87 mg/dL, *p* < 0.05). The AG correlated positively with creatinine (r = 0.44, *p* < 0.01) and urea (r = 0.53, *p* < 0.01), but there was no correlation between HCO_3_^−^ and both creatinine (r = −0.015, *p* > 0.05) and urea (r = −0.07, *p* > 0.05). The Cl^−^ concentrations correlated negatively with HCO_3_^−^ (r = −0.8, *p* < 0.01). Conclusions: The most common type of acid–base disturbance in CKD patients in stages 4 and 5 is AGMA, which is observed in patients with better kidney function and is associated with compensatory hyperchloremia. The initiation of renal replacement therapy was significantly earlier for patients diagnosed with HAGMA compared to those diagnosed with AGMA. The more advanced the CKD, the higher the AG.

## 1. Introduction

Metabolic acidosis (MA) is a common disorder in patients with chronic kidney disease (CKD), but even in advanced stages, it may not be observed. MA affects the development of a number of complications, including protein malnutrition, increased inflammation, muscle wasting, hypoalbuminemia and the development of metabolic bone disease, and is responsible for an increase in mortality among CKD patients and a glomerular filtration rate (GFR) decline. The kidneys play a pivotal role in the acid–base balance and maintain pH in a normal range (isohydria) through three mechanisms: bicarbonate reabsorption, ammogenesis and titratable acidity. In conditions of renal failure, endogenous acids accumulate, and bicarbonate is used to buffer them. A bicarbonate level below 22 mmol/L is the generally accepted diagnostic criterion for metabolic acidosis. In CKD, acidosis can be associated with a normal anion gap (AGMA) and a high anion gap (HAGMA). The anion gap is the difference between the serum concentration of cations and anions and is usually calculated using the formula AG = Na^+^ − (Cl^−^ + HCO_3_^−^), although other particles like potassium (cation), phosphate (anion), albumin (anion), calcium (cation) and magnesium (cation) may affect it. In renal failure, an accumulation of anions such as sulfone derivatives (indoxyl, p-cresol) is observed, which may increase AG. In AGMA, bicarbonate depletion is compensated by an increase in the concentration of the anion Cl^−^. The traditional division of metabolic acidosis into HAGMA and AGMA assumes a complete compensation of the anion gap in AGMA by hyperchloremia and no change in chloride concentration in cases with HAGMA [1,2].

When metabolic acidosis is caused by, for example, a loss of bicarbonate, hyperchloremic acidosis develops with a normal AG. There are numerous mechanisms through which bicarbonate loss can result in hyperchloremic metabolic acidosis, including renal, gastrointestinal and exogenous causes. In these cases, the negatively charged chloride ion enters the extracellular space due to the loss of bicarbonate, and a normal AG is maintained. The most frequent causes of HAGMA include glycols (ethylene, propylene and diethylene), 5-oxoproline (acetaminophen), L-lactic acid, D-lactic acid, methanol, aspirin, renal failure and ketoacidosis. They are arranged under the mnemonic “GOLDMARK”. In all these disorders, metabolic acidosis reduces the amount of HCO_3_^−^ without changing the amount of Cl^−^ and thereby increases the AG in a reciprocal fashion. [1].

In this paper, we test whether this general rule applies to patients with CKD in advanced stages and analyze the relationship between point-of-care testing (POCT) parameters and the renal function.

The main aim of the study is to assess the prevalence of various types of acidosis because most of the literature presents data from the whole spectrum of patients with CKD; therefore, in this manuscript, we focused on patients with advanced CKD (stages IV and V) [3]. Finally, we assessed the clinical status of each participant at a 2-year follow-up depending on the type of metabolic acidosis.

In the study, we used a new method of collecting arterial blood for POCT directly from the artery during the creation of an arteriovenous fistula.

## 2. Materials and Methods

Initially, 188 patients with CKD in stage G4 or G5, admitted to our department to create an arteriovenous fistula, were screened for participation in the study from January 2020 to June 2021. The inclusion criteria were: (1) aged over 18 y old; (2) patients with CKD stages G4 and G5 who qualified for fistula creation but were not on a hemodialysis program; 3) signed a written consent. In order to enhance the homogeneity of the cohort, the decision was made to exclude patients undergoing hemodialysis. The bicarbonate concentrations in dialysis fluid are high (typically 32 mmol/L), resulting in a quick correction of metabolic acidosis during hemodialysis. The heterogeneous study group, which included HD and non-HD patients, may have complicated the results analysis. In total, 72 patients were excluded due to their already ongoing hemodialysis treatment. The other 116 patients (74 male, 42 female) were recruited for this cross-sectional study. The participants received no additional dietary recommendations other than those provided by the main physician during follow-up visits in the out-patient clinic. The nutrition/malnutrition status was checked in all participants using the Nutritional Risk Score 2002 (NRS), Subjective Global Assessment, SGA, and serum albumin concentration. The NRS is a short screening tool that assesses the risk of malnutrition in hospitalized patients. It consists of two parts: a nutritional screening section and a nutritional assessment section. The screening section includes questions about weight loss, body mass index and food intake. The assessment section includes questions about disease severity, age, and dietary intake. Regarding the SGA, it is a clinical tool that assesses the nutritional status of patients based on their medical history, a physical examination, and laboratory data. It is used to identify patients who are at risk of malnutrition and was validated in CKD stage 4–5 patients. Both tools were used upon admission to hospital, and none of the patients showed malnutrition.

Four patients within the study group were receiving bicarbonate. They were instructed to discontinue its use three days prior to surgery.

A collection of arterial blood samples was conducted during the procedure of arteriovenous fistula (AVF) creation. After dissecting the radial artery (RA) and clamping the distal and proximal sections, a longitudinal incision was made. Blood was then drawn into a heprinized syringe after the clamp was released using a plastic needle inserted into the proximal part of the RA. The time between RA closure and sample collection was no longer than 30 s. The assisting nurse then took the sample to the POCT in an adjacent room and immediately performed the test. The total time from sample collection to measurement did not exceed 3 min. An analyzer (RADIOMETER ABL90 SERIES, RADIOMETER MEDICAL APS, Denmark) was used to examine POCT parameters. This is a fully automatic device and is used to obtain diagnostic results while with the patient or close to the patient. It has a cartridge containing test reagents and performs an 8 h calibration. The following parameters were assessed: pCO_2_, carbon dioxide partial pressure; HCO_3_^−^, bicarbonate; ABE, actual base excess; SBE, standard base excess; K^+^, potassium; Na^+^, sodium; Ca^2+^, ionized calcium; Cl^−^, chloride; AG, anion gap; and AG (K^+^), adjusted anion gap for the contribution of serum potassium, urea and creatinine. An estimated glomerular filtration rate (eGFR) was calculated based on the MDRD formula as our device did not report it automatically [4].

Table 1 presents the main demographic and clinical parameters of the patients included in the study.

Next, this group of 116 patients was divided into three subgroups: (1) patients without metabolic acidosis (non-MA) with HCO_3_^−^ > 22 mmol/L, (2) patients with MA with a normal anion gap (AGMA) (AG ≤ 10 mmol/L, HCO_3_^−^ ≤ 22 mmol/L) and (3) patients with high-anion-gab MA (HAGMA) (AG > 10 mmol/L, HCO_3_^−^ ≤ 22 mmol/L). 

The primary causes of chronic kidney disease (CKD), as well as demographic and comorbidity data, were collected from medical records and direct interviews. The Charlson comorbidity index (CCI) was counted according to the rule presented in precious study [5,6]. Briefly, this score includes the patient’s age and the following diseases listed in groups: (1) heart diseases; myocardial infarction (MI) with a history of definite or probable MI (EKG changes and/or enzyme changes); chronic heart failure with exertional or paroxysmal nocturnal dyspnea, and the patient has responded to digitalis, diuretics or afterload-reducing agents; (2) peripheral obstructive arterial disease (PAOD) with intermittent claudication or past bypass for chronic arterial insufficiency, a history of gangrene or acute arterial insufficiency, or untreated thoracic or abdominal aneurysm (≥6 cm); (3) cerebrovascular accident (CVA) or transient ischemic attacks (TIA) with minor or no residual deficit; dementia with chronic cognitive deficit; hemiplegia (4) chronic obturative pulmonary disease (COPD); (5) connective tissue disease; (6) gastrointestinal diseases: peptic ulcer disease with any history of treatment for ulcer disease or history of ulcer bleeding; liver disease; (7) diabetes mellitus; (8) neoplasmatic disease: solid tumor; leukemia; lymphoma. Depending on the severity of the comorbidity, 1–6 points were assigned, and the sum was calculated.

A two-year clinical follow-up was conducted to assess the clinical status of each participant by reviewing their medical records and through a telephone interview. All important information has been recorded, including the time of dialysis initiation, cardiovascular events, deaths, and kidney transplants.

In addition, ambulatory measurements of hemodynamic parameters were taken with a Mobil-O-Graph monitor (Industrielle Entwicklung Medizintechnik und Vertriebsgesellschaft GmbH -IEM, Stolberg, Germany), which records oscillometric arm blood pressure: systolic and diastolic blood pressure (SBP and DBP), pulse pressure (pPP), ejection fraction (EF), cardiac output (CO) and pulse waves. It calculates the pulse wave velocity (PWV) as a measure of arterial stiffness. 

A statistical analysis was performed using standard software (Statistica Version 13.3, StatSoft, Tulsa, OK, USA). Continuous variables between groups were expressed as a mean and standard deviation (±SD) and compared using the independent t-test, an analysis of variance (ANOVA), the Mann–Whitney U test or the Kruskal–Wallis ANOVA test, based on the number of groups, to compare the normality of the variables, tested using the Kolmogorov–Smirnov test. Categorical variables were expressed as an absolute number (n) and a percentage (%) and compared using the χ^2^ test or Kruskal–Wallis ANOVA test. The relationship between the POCT parameters was examined using Pearson’s correlation analysis. A *p*-value less than 0.05 was considered significant.

Ethics approval was granted by the Ethics Board of Wroclaw Medical University (No. KB-609/2019).

## 3. Results

The study included 116 non-dialysis patients with chronic kidney disease in stages G4 and G5, with a mean age of 62.5 ± 17 years. The general demographic, clinical data, primary cause of chronic kidney disease (CKD) and laboratory characteristics of these patients are presented in Table 1. The entire group of 116 patients was divided into three main subgroups according to the criteria provided in the Section 2. The number of patients with non-MA, AGMA and HAGMA was 24 (21%), 48 (41%) and 44 (38%), respectively. Based on the ANOVA test, significant intergroup differences were found (Table 1). In both subgroups with metabolic acidosis, i.e., AGMA and HAGMA, significantly higher concentrations of Cl^−^ and lower pHs and concentrations of pCO2 and HCO3^−^ were observed compared to the non-MA group. The anion gap (AG) was the highest in the HAGMA group and the lowest in the AGMA group. The HAGMA group was characterized by more advanced renal failure and the lowest estimated glomerular filtration rate (eGFR). The albumin concentrations were statistically significantly lower in the HAGMA group. No significant association was found between the type of acid–base disorder and the cause of CKD.

The post hoc analyses showed significant intergroup differences, which are depicted in Table 1. The bicarbonate levels were significantly higher in patients without acidosis (non-MA) in comparison to both the AGMA and HAGMA subgroups. The opposite relation applied to Cl^−^, where its concentration was significantly lower in the subgroups of patients with both acidosis (AGMA and HAGMA). The pH, pCO_2_, actual and standard base excesses were higher in the non-MA group compared to the AGMA and HAGMA groups. The lowest albumin concentration was observed in the HAGMA patients. The most advanced renal failure was observed in the HAGMA group, while no significant differences were observed between the non-MA and AGMA groups.

In the entire group of 116 patients, decompensated acidosis with a pH < 7.35 was found in 68 (59%) patients. This group is characterized by a statistically significantly lower eGFR, a lower concentration of bicarbonate and lower base excess, and a higher concentration of chloride and phosphate (Table 2). There was no correlation between the underlying etiology of CKD and the degree of acidosis control.

For follow-up purposes, the following data were collected: the time of dialysis initiation, cardiovascular events, deaths, and kidney transplantations (Table 3). The follow-up time ranged from 15 to 51 months (with a mean of 31 months). The time interval between fistula creation (i.e., taking a POCT sample) and starting dialysis treatment was 3–913 days (mean 187 days). Thirteen patients did not start renal replacement therapy, including four who were lost to follow-up. During the follow-up period, 102 patients started dialysis treatment. Among the 116 study patients, there were 23 deaths. This was found to be associated with cardiovascular events in twelve cases, infectious complications in eight cases, and cancer in three cases. A kidney transplantation was performed in 15 patients. Cardiovascular complications occurred in 30 patients and included myocardial infarction, the need for coronary angioplasty or coronary revascularization, paroxysmal atrial fibrillation, decompensation of heart failure and ischemic and hemorrhagic stroke. We found that patients with HAGMA have a significantly shorter time to start renal replacement therapy in comparison to the AGMA group (113 ± 184 vs. 238 ± 229 days, *p* < 0.05) (Table 3). There were no other intergroup differences for clinical data in the three metabolic acidosis subgroups (non-MA vs. AGMA vs. HAGMA) (Table 3). There were also no differences between the two pH subgroups (pH < 7.35 vs. pH ≥ 7.35) (Table 4). 

A strong negative correlation was shown between the bicarbonate and chloride concentrations (Figure 1) and a positive correlation between the anion gap and renal function parameters (creatinine and urea) (Figure 2 and Figure 3). There were no statistically significant correlations between the bicarbonate and creatinine (r = 0.12, *p* > 0.05) or urea (r = 0.07, *p* > 0.05) concentrations. There were also no significant correlations between the chloride and creatinine (r = 0.12, *p* > 0.05) or urea (r = 0.10, *p* > 0.05) concentrations.

## 4. Discussion

Our population of patients with CKD in stages 4 and 5 was characterized by considerable diversity in the context of acid–base balance (ABB) disorders. First, 21% of the patients did not show metabolic acidosis (MA), defined as a bicarbonate concentration of less than 22 mmol/L. In this group, the ABB disturbances were the least significant, which we assume is related to the preserved function of the renal tubules for (1) HCO_3_^−^ reabsorption, (2) proton (H^+^) secretion by titrated acid excretion (TAE) and/or (3) ammonium (NH4^+^) production (renal ammoniagenesis), despite the globally low glomerular filtration rate (GFR). In the second group of patients with normal-anion-gap metabolic acidosis (AGMA), we observed impaired acidosis correction with lower bicarbonate concentrations, but the anion gap was maintained at a normal level due to a compensatory resorption of chloride (Cl^−^). This was the largest group, comprising 41% of the patients, in which we found the smallest accumulations of nitrogen metabolites with the lowest creatinine and urea concentrations. In the last subgroup of patients with high-anion-gap metabolic acidosis (HAGMA), both problems were observed, i.e., HCO_3_^−^ resorption disorders and increased AG as a result of unmeasured anion accumulation due to the dysfunction of all the other mentioned processes responsible for balancing acidosis. This type of MA accounted for 38% of all the study patients in whom hyperchloremia was unable to compensate for the anion gap. This was the subgroup with the lowest eGFR. 

Based on these data, it appears that the three above-mentioned processes responsible for the acid–base balance have different importance at different stages of the CKD trajectory [7]. In the earlier stages of CKD, the serum bicarbonate concentration is usually within the normal range, although acid accumulation occurs and the positive acid balance is buffered by bone and tissue without affecting plasma HCO_3_^−^ concentrations. This is often referred to as “hidden acidosis”. In this setting, ammonium excretion is reduced; however, by maintaining the renal removal of phosphate and other anions, as well as hyperchloremia, a normal anion gap is maintained [8]. It should be pointed out that ammonium (NH4^+^) is secreted by the distal nephron in conjunction with the chloride anion [9]. Although our population included only patients in the late stages of CKD, we found an inverse, strong relationship between bicarbonate and chloride concentrations (Figure 1). A similar relationship between chloride and bicarbonate is observed in renal tubular acidosis (RTA), where impaired bicarbonate reabsorption prevents normal increases in luminal chloride concentration, so RTA may be a model of acid–base balance disorders in CKD. It has long been suspected that tubular dysfunction and impaired tubular acid excretion dominate the global loss of renal function and contribute to hyperchloremia [10,11]. It has been confirmed that in patients with any kidney disease, the risk of acidosis is 1.7 times higher with a diagnosis of tubulointerstitial nephritis [12]. Our results do not support this, and no significant association was found between the type of acid–base disorder and the primary cause of CKD. It is likely that in advanced CKD stages 4 and 5, the function of the renal tubules most involved in acidosis compensation is similarly impaired, regardless of the underlying renal disease. It should be pointed out that there was no correlation between creatinine and both chloride and bicarbonate concentrations. We suspect that this is due to the accumulation of study participants with a restricted GFR, i.e., a low GFR, and we believe that in a larger cohort of patients, including those with CKD at earlier stages, we would observe more typical relationships.

Bicarbonate is mainly (in 85–90% of the filtered bicarbonate) resorbed in the proximal tubule. The loop of Henle reabsorbs around 10%, and the remaining 5–10% is reabsorbed in the collecting tubules [13]. Different types of carbonic anhydrase (CA) are involved in this process, namely membrane and cytoplasmic CA. In the proximal tubule, the filtered bicarbonate reacts with hydrogen ions to form carbonic acid (H_2_CO_3_), which is broken down into CO_2_ and H_2_O by the membrane-bound carbonic anhydrase IV (CA IV) enzyme. CO_2_ diffuses into tubular cells, where it reacts with H_2_O, and in the presence of cytoplasmic carbonic anhydrase II (CA II), it recombines with HCO_3_^−^ and H^+^. Next, HCO_3_^−^ is transported to the blood in exchange for chlorides. The latter process of reversible electroneutral exchange is carried out by the Cl^−^/HCO_3_^−^ exchanger and allows the anion gap to be kept within the correct range [14]. These physiological processes indirectly explain the interaction between Cl^−^ and HCO_3_^−^ in CKD, as demonstrated in Figure 1.

As CKD progresses to more advanced stages, the serum bicarbonate concentration gradually decreases, which is associated with compensation for acidosis and abnormal renal resorption. Animal experiments indicate that a reduced amount of HCO_3_^−^ is absorbed by the tubules, and this is accompanied by an impaired secretion of Cl^−^ into the tubules, which causes an increase its concentration in the serum [15]. As mentioned above, this mechanism is sufficient to maintain a normal AG at an earlier stage of chronic kidney disease. However, this process fails as kidney function deteriorates, and HAGMA develops. This group is characterized by a high AG and high chloride concentration. This type of MA differs from the classic HAGMA caused by other causes, including glycols (ethylene, propylene and diethylene), 5-Oxoproline (acetaminophen), L-lactic acid, D-lactic acid, methanol, aspirin and ketoacidosis, because, as a rule, changes in chloremia are not observed in classical variants of HAGMA [1].

The serum anion gap (AG) is calculated as the difference between sodium (main cation) and major anion (bicarbonate and chloride) concentrations. The AG is affected by both measurable anions (albumin, phosphates) and unmeasurable anions, including those accumulated in kidney disease meltabilities, such as indoxyl, p-cresol and other uremic solutes [16]. Typically, for CKD, changes in the concentrations of measurable cations (potassium, magnesium, calcium) may also affect the level of the AG. We found a positive correlation between the AG and creatinine and urea concentrations (Figure 2 and Figure 3). This finding is consistent with the observations of others. The NHANTES data show a gradual increase in the albumin-corrected AG across eGFR categories, starting at 45–59 mL/min/1.73 m^2^ [17]. In another study, the AG was a marker for the retention of various solutes and a measure of renal tubular function, which, according to the authors, may provide a more sensitive measure of impaired renal function [17]. Elevated levels of uremic toxins (indoxyl sulfate and p-cresol sulfate) have been shown in patients with chronic kidney disease; however, it should be emphasized that the concentration of uremic toxins did not correlate with the GFR, possibly due to tubular secretion and reabsorption, which also play a role in the clearance of uremic toxins [18,19,20]. In addition, the net number of unmeasured anions appears to be an independent predictor of CKD progression and plays an important role in the development of CKD complications [17].

According to pH, we additionally divided the entire study group into two subgroups, i.e., those with decompensated and compensated acidosis. The group of patients with decompensated acidosis (pH < 7.35) was characterized by more advanced ABB disorders (lower bicarbonate concentration and lower base excess) and higher creatinine concentrations in comparison to patients with a pH ≥ 7.35. In other words, along with the progression of CKD and a decrease in the GFR, deeper disturbances of renal tubular function were observed, including the ability to compensate for metabolic acidosis, which is reflected in changes in basic acid–base parameters (bicarbonate, BE—see Table 2). These changes also affect the phosphaturic function of the kidneys. The patients with decompensated acidosis had higher levels of phosphate and parathyroid hormone, i.e., more advanced secondary hyperparathyroidism. It is noteworthy that hyperparathyroidism, defined as PTH > 60 pg/mL, was present in nearly 20% of patients with stage 2 CKD and appeared much earlier in CKD than other complications such as anemia, acidosis, hyperkaliemia and hyperphosphatemia [12]. On the other hand, some authors suggest that metabolic acidosis is associated with the exacerbation of secondary hyperparathyroidism, and its correction may have a beneficial effect [21].

Among the participants in the third National Health and Nutrition Examination Survey (NHANTES), 19% of the subjects with GFR 15–29 mL/min/1.73 m^2^ (CKD stage G4) had a serum bicarbonate concentration below 22 mmol/L [3]. By comparison, in our study, which included patients with CKD in stages G4 and G5, metabolic acidosis (MA) with a bicarbonate concentration less than 22 mmol/L was found in 92 out of 116 (79%) patients. This difference was probably due to a broader population, which included patients with more advanced CKD. Additionally, the participants received no additional dietary recommendations other than those provided by the main physician during the follow-up visits in the out-patient clinic, and this could explain the increased prevalence of acidosis. This study demonstrated that metabolic acidosis is a neglected area of nephrology that requires improvement. In another single-center study, the incidence of metabolic acidosis in patients with stage G5 and stage G4 CKD was 56% and 38%, respectively [22]. 

The distinction between different types of ABB disorders in advanced CKD may be important in the context of HCO_3_^−^ supplementation. We hypothesized that bicarbonate may be less effective in normal AGMA compared to HAGMA due to the more profound renal tubular dysfunction in the latter group. This should be the subject of future research. There are studies indicating that in patients with CKD in stages G3 and G4, bicarbonate supplementation may not be effective and does not lead to an improvement in the parameters of secondary hyperparathyroidism (bone mineral density and muscle function after one year) [23].

It should be noted that in our study, bicarbonate concentration did not correlate with renal function. It is generally known that its concentration is related to diet. According to data from NHANTES, an inverse relationship was found between serum bicarbonate and dietary acid in middle-aged and older adults. In young participants, serum bicarbonate levels did not differ with different acid loads, suggesting that they could handle higher acid loads without altering the acid–base balance [24]. In addition, intentionally reducing protein intake can lead to an increase in tCO2 [25]. 

Furthermore, a two-year clinical follow-up was performed. It was found that patients assigned to the HAGMA group started renal replacement therapy considerably earlier than those in the AGMA group. The practical significance of this conclusion stems from the fact that if a patient is classified as having HAGMA, the nephrologist is obligated to speed up making plans for renal replacement therapy, including the creation of a vascular access. Naturally, this finding should be interpreted with caution because the criterion for starting dialysis is multifactorial and the decision is influenced by clinical features such as overhydration, high nitrogen metabolite values, hyperpotassemia, etc. Further research should address this topic.

One of the important elements of an assessment of a patient with CKD is the assessment of acid–base disorders. In our study, we collected arterial blood during AVF creation since it identifies such disorders more precisely than venous blood. To our knowledge, this type of diagnostic has never been used for POCT collection. It has the advantage that it is not associated with arterial trauma, e.g., hematoma or other complications, but it provides important information regarding acid–base disorders.

The main limitation of the study is that it is cross-sectional; thus, we were unable to observe changes in the POCT values over time, but we did conduct a clinical 2-year follow-up with interesting results, as reported above. We did not measure protein-bound uremic solutes because this would replicate other well-known studies on this topic. As is well-known, most patients with chronic kidney disease have metabolic acidosis and a normal serum anion gap; an increased anion gap is uncommon unless the eGFR is extremely low (i.e., less than 15 mL/min/1.73 m^2^) due to phosphate, sulfate and other anion accumulations [26]. We did not test for protein-bound uremic toxins, for example, indoxyl sulfate, because this is not a standard parameter that is regularly assessed. Chromatography is used to measure serum total and free indoxyl sulfate as well as p-cresyl sulfate; however, this technique is not easily accessible. Our study concentrated mainly on POCT parameters and basic biochemical laboratory indices that are readily accessible to most practitioners. We also did not measure any urinary indices, including urinary ammonia, titratable acid excretion and the urinary anion gap, to correlate with serum AG, and we did not correlate our results with the patients’ diet. The participants in our study received no additional dietary recommendations other than those provided by the main physician during the follow-up visits in the out-patient clinic, and this could explain the increased prevalence of acidosis. Furthermore, arterial blood was used for the POCT, whereas venous blood is used in the majority of acid–base studies. Some studies indicate that these differences may be significant [27]. 

## 5. Conclusions

The most common type of acid–base disturbance in CKD patients in stages 4 and 5 is AGMA, which is observed in patients with better kidney functions and is associated with compensatory hyperchloremia.

Our study demonstrated that in patients with advanced CKD and HAGMA, hyperchloremia is also observed, which differs from the traditional concept [1] that chloride concentration is normal in this type of MA. Furthermore, our observations are consistent with previous studies that documented an increase in serum chloride concentration in the early stages of CKD, while the anion gap remained normal [27]. In more advanced CKD, a gradual increase in the anion gap is observed, while the chloride concentration is usually still elevated. On the contrary, we found a negative correlation between Cl^−^ and HCO_3_^−^, concentrations, which reflects typical compensatory changes in CKD targeted at maintaining a normal AG. In contrast to classical HAGMA, characterized by high AG values without changing Cl^−^ concentrations, in the population with advanced CKD, the level of chlorides is usually elevated in this type of MA. 

A positive correlation was found between the AG and eGFR, and it was found that HAGMA patients require the initiation of dialysis treatment earlier than AGMA patients. 

These observations shed light on the evolving nature of acid–base disturbances in CKD, highlighting the importance of monitoring both the anion gap and chloride levels in advanced CKD patients for a comprehensive understanding of their metabolic acid–base status.

## Figures and Tables

**Figure 1 diagnostics-13-03367-f001:**
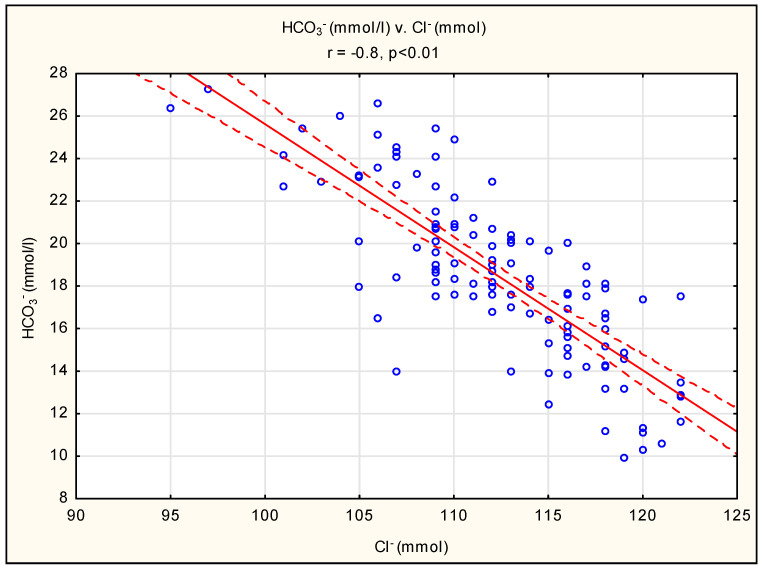
Pearson’s correlation between bicarbonate and chloride concentrations. The red line describes the linear correlation line. The red-dashed line is the standard deviation line. A blue circle represents each case of study.

**Figure 2 diagnostics-13-03367-f002:**
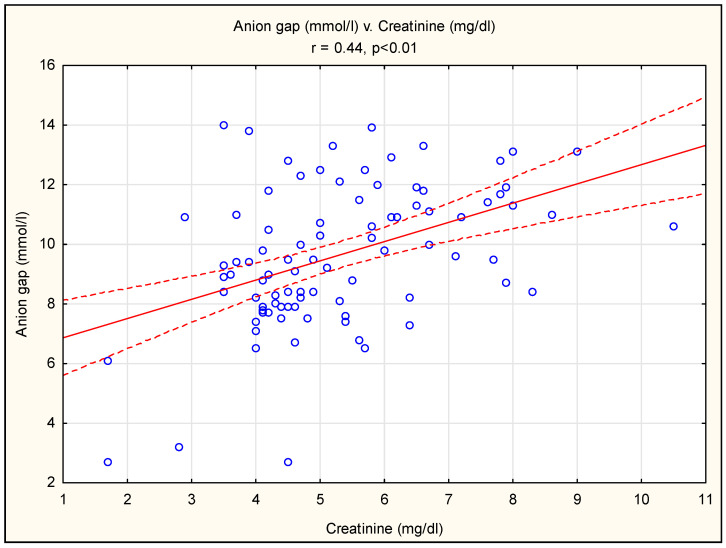
Pearson’s correlation between anion gap and creatinine concentration. The red line describes the linear correlation line. The red-dashed line is the standard deviation line. A blue circle represents each case of study.

**Figure 3 diagnostics-13-03367-f003:**
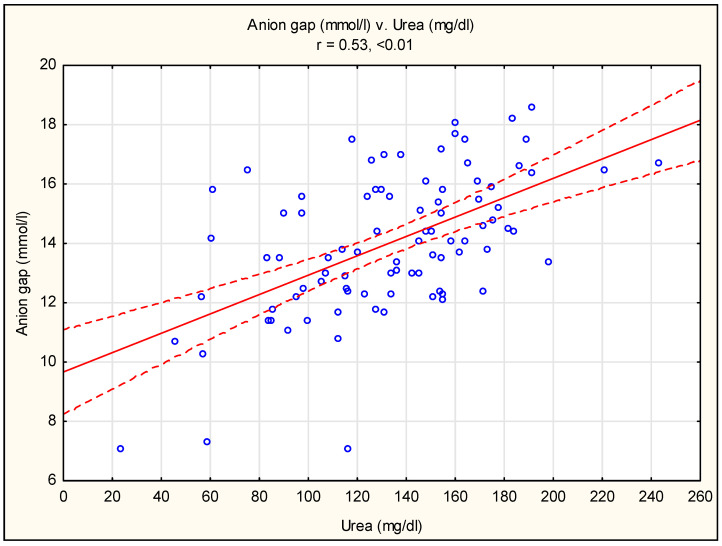
Pearson’s correlation between anion gap and urea concentration. The red line describes the linear correlation line. The red-dashed line is the standard deviation line. A blue circle represents each case of study.

**Table 1 diagnostics-13-03367-t001:** General demographic, clinical and laboratory characteristics of the study group divided into three main subgroups.

	non-MA *N* = 24	AGMA *N* = 48	HAGMA *N* = 44	F	*p* *	All Study Patients *N* = 116
Age (y.)	63.52 ± 14.78	62.48 ± 16.88	62.02 ± 18.5	0.06	>0.05	62.52 ± 16.98
BMI (kg/m^2^)	28.62 ± 6.61	26.63 ± 4.61	26.3 ± 4.75	1.64	>0.05	26.92 ± 5.17
pH	7.39 ± 0.03 ^a^	7.32 ± 0.05 ^b^	7.31 ± 0.06 ^b^	25.36	<0.01	7.33 ± 0.06
pCO_2_ (mmHg)	39.88 ± 2.86 ^a^	33.44 ± 4.94 ^b^	32.19 ± 4.38 ^b^	25.66	<0.01	34.30 ± 5.22
pO_2_ (mmHg)	78.19 ± 13.97 ^a^	83.29 ± 13.25 ^a^	91.3 ± 19.6 ^b^	5.78	<0.05	85.27 ± 16.75
HCO_3_^−^ (mmol/L)	24.20 ± 1.41 ^a^	17.51 ± 2.97 ^b^	16.23 ± 2.89 ^b^	72.79	<0.01	18.41 ± 4.03
ABE (mmol/L)	−0.51 ± 1.64 ^a^	−7.64 ± 3.27 ^b^	−9.07 ± 3.28 ^b^	66.56	<0.01	−6.70 ± 4.4
SBE (mmol/L)	−0.53 ± 1.74 ^a^	−8.54 ± 3.54 ^b^	−10.06 ± 3.55 ^b^	71.05	<0.01	−7.46 ± 4.85
sO_2_ (%)	94.90 ± 3.76	95.49 ± 3.6	96.23 ± 3.37	1.17	>0.05	95.65 ± 3.55
K^+^ (mmol/L)	4.23 ± 0.98	4.48 ± 0.7	4.41 ± 0.77	0.76	>0.05	4.4 ± 0.79
Na^+^ (mmol/L)	139.54 ± 3.82 ^a^	139.81 ± 3.75 ^a^	141.68 ± 2.02 ^b^	5.11	<0.05	140.47 ± 3.33
Ca^2+^ (mmol/L)	1.11 ± 0.14	1.16 ± 0.09	1.13 ± 0.1	1.27	>0.05	1.13 ± 0.11
Cl^−^ (mmol/L)	105.38 ± 4.17 ^a^	113.56 ± 5.69 ^b^	114.23 ± 4.03 ^b^	29.90	<0.01	112.12 ± 5.9
Anion gap (mmol/L)	9.82 ± 2.81 ^a^	8.21 ± 1.72 ^b^	11.31 ± 1,72 ^c^	27.87	<0.01	9.72 ± 2.41
Anion gap (K^+^) (mmol/L)	14.06 ± 2.83 ^a^	12.72 ± 1.77 ^b^	15.71 ± 1.96 ^c^	23.29	<0.01	14.13 ± 2.47
Glucose (mg/dL)	137.67 ± 61.69	116.58 ± 41.05	111.22 ± 38	2.79	>0.05	118.91 ± 45.7
TSH (iuU/mL)	1.94 ± 1.62	1.72 ± 1.21	1.87 ± 1.34	0.20	>0.05	1.82 ± 1.35
Total protein (g/dL)	5.9 ± 1.13	6.1 ± 0.82	5.79 ± 0.95	1.35	>0.05	5.94 ± 0.94
Albumin (g/dL)	3.22 ± 0.73	3.56 ± 0.48 ^a^	3.2 ± 0.63 ^b^	4.83	<0.05	3.35 ± 0.61
TC (mg/dL)	208.45 ± 74.57	183.49 ± 63.58	189.53 ± 69.5	0.99	>0.05	191.02 ± 68.23
TG (mg/dL)	180.76 ± 111.45	143.16 ± 76.58	156.28 ± 66.25	1.51	>0.05	155.8 ± 81.6
CRP (mg/L)	6.54 ± 11.93	11.29 ± 19.17	7.7 ± 9.64	1.07	>0.05	8.96 ± 14.77
Pi (mg/dL)	5.55 ± 1.45	5.21 ± 1.06	5.67 ± 1.42	1.57	>0.05	5.46 ± 1.29
Alkaline phosphatase (IU/L)	84.62 ± 33.27	71.92 ± 23.49	72.85 ± 33.58	1.24	>0.05	74.40 ± 29.54
BNP (pg/mL)	500.54 ± 749.09	486.87 ± 510.89	696.52 ± 1050.93	0.56	>0.05	555.54 ± 768.88
Troponin I (ng/L)	127.28 ± 249.58	138.34 ± 367.99	85.54 ± 207.12	0.18	>0.05	117.31 ± 292.96
Hemoglobin (g/dL)	10.16 ± 1.21	9.97 ± 1.44	9.99 ± 1,63	0.14	>0.05	10.02 ± 1.46
Creatinine (mg/dL)	5.65 ± 2.43	4.75 ± 1.15 ^a^	5.99 ± 1.95 ^b^	5.86	<0.01	5.39 ± 1.85
eGFR (mL/min/1.73 m^2^)	12.21 ± 4.22	12.73 ± 4.86 ^a^	9.97 ± 2.78 ^b^	5.49	<0.01	11.61 ± 4.23
Urea (mg/dL)	146.33 ± 47.61	131.83 ± 35.91 ^a^	164.97 ± 39.19 ^b^	7.86	<0.05	146.96 ± 42.08
PWV (m/s)	8.34 ± 1.81	10.87 ± 2.22	9.77 ± 3.02	1.79	>0.05	9.9 ± 2.5
Smoking (pack-year)	12.71 ± 17.86	8.92 ± 14.32	9.15 ± 14.49	0.55	>0.05	9.84 ± 15.15
Comorbidities ***						
	non-MA *N* = 24	AGMA *N* = 48	HAGMA *N* = 44	F	*p* *	All study patients *N* = 116
CCI (points)	5.83 ± 2.78	6.35 ± 3.23	6.77 ± 3.13	0.72	>0.05	6.4 ± 3.1
	non-MA *N* = 24	AGMA *N* = 48	HAGMA *N* = 44	H	*p* **	No (%)
Heart diseases (%)	12 (10)	19 (16)	15 (13)	1.34	>0.05	46 (40)
Peripheral vascular disease (%)	9 (8)	10 (9)	8 (7)	5.54	>0.05	27 (23)
Cerebrovascular accident (%)	6 (5)	7 (6)	10 (9)	3.45	>0.05	23 (20)
Chronic obturative disease (%)	2 (2)	4 (3)	4 (3)	0.05	>0.05	10 (9)
Connective tissue disease (%)	2 (2)	6 (5)	4 (3)	0.31	>0.05	12 (10)
Gastrointerstitial diseases (%)	4 (3)	7 (6)	5 (4)	0.29	>0.05	16 (14)
Diabetes mellitus (%)	11 (9)	15 (13)	13 (11)	1.91	>0.05	39 (34)
Neoplasmatic disease (%)	2 (2)	7 (6)	5 (4)	0.53	>0.05	14 (12)
Causes of CKD						
	non-MA N = 24	AGMA *N* = 48	HAGMA *N* = 44	H	*p* **	No (%)
DM and HA (%)	14 (58)	24 (50)	18 (41)	2.94	>0.05	56 (48)
Chronic GN (%)	8 (33)	10 (21)	13 (30)			31 (27)
ADPKD (%)	0 (0)	7 (15)	5 (11)			12 (10)
IN (%)	0 (0)	4 (8)	1 (2)			5 (4)
others (%)	2 (0)	3 (6)	7 (16)			12 (10)

Mean ± standard deviation. * ANOVA with Tukey’s post hoc test. Different letters in rows imply statistical differences at the significance level *p* < 0.05. ** Kruskal–Wallis ANOVA test, *p*-value < 0.05 statistically significant. Abbreviations: non-MA, non-metabolic acidosis; AGMA, metabolic acidosis with normal anion gab; HAGMA, high-anion-gab metabolic acidosis; BMI, body mass index; pO_2_, oxygen partial pressure; pCO_2_, carbon dioxide partial pressure; HCO_3_^−^, bicarbonate; ABE, actual base excess; SBE, standard base excess; K^+^, potassium; Na^+^, sodium; Ca^2+^, ionized calcium; Cl^−^, chloride; TSH, Thyroid Stimulating Hormone; TC, total cholesterol; TG, Triglycerides; CRP, C Reactive Protein; Pi, phosphate; BNP, B-type natriuretic peptide; eGFR, estimated glomerular filtration rate; PWV, pulse wave velocity; CCI, Charlson comorbidity index; CKD, chronic kidney disease; DM, diabetes mellitus; HA, hypertension; GN, glomerulonephritis; ADPKD, autosomal dominant polycystic kidney disease; IN, interstitial nephritis. *** Comorbidities are listed in groups in the Material and Methods section.

**Table 2 diagnostics-13-03367-t002:** Comparison of clinical and laboratory parameters in two subgroups of patients with compensated (pH > 7.35) and decompensated (pH ≥ 7.35) acidosis.

	pH < 7.35 *N* = 68	pH ≥ 7.35 *N* = 48	*p*
Age (y.)	63.07 ± 17.79	61.75 ± 15.93	>0.05
Weight (kg)	77.33 ± 17.13	77.84 ± 22.56	>0.05
BMI (kg/m^2^)	27.35 ± 4.79	26.33 ± 5.65	>0.05
pH	7.29 ± 0.04	7.39 ± 0.03	<0.01
pCO_2_ (mmHg)	33.52 ± 5.54	35.39 ± 4.57	>0.05
pO_2_ (mmHg)	86.16 ± 17.51	84.06 ± 15.75	>0.05
HCO_3_^−^ (mmol/L)	16.39 ± 3.36	21.27 ± 3.08	<0.01
ABE (mmol/L)	−9.24 ± 3.45	−3.12 ± 2.86	<0.01
SBE (mmol/L)	−10.14 ± 3.84	−3.66 ± 3.37	<0.01
K^+^ (mmol/L)	4.51 ± 0.73	4.25 ± 0.85	>0.05
Na^+^ (mmol/L)	139.77 ± 3.99	141.13 ± 2.28	<0.05
Ca^2+^ (mmol/L)	1.14 ± 0.1	1.13 ± 0.12	>0.05
Cl^−^ (mg/dL)	114.69 ± 5.09	108.52 ± 4.95	<0.01
Anion gap (mmol/L)	9.54 ± 2.54	9.97 ± 2.22	>0.05
Anion gap (K^+^) (mmol/L)	14.06 ± 2.61	14.22 ± 2.28	>0.05
Creatinine (mg/dL)	5.46 ± 1.61	5.06 ± 1.61	>0.05
GFR (ml/min/1.73 m^2^)	9.25 ± 2.73	10.2 ± 3.24	<0.05
Urea (mg/dL)	135.14 ± 33.58	129.64 ± 49.43	>0.05
TSH (iuU/mL)	1.9 ± 1.52	1.72 ± 1.08	>0.05
Total protein (g/dL)	5.76 ± 0.83	6.19 ± 1.04	<0.05
Albumin (g/dL)	3.35 ± 0.62	3.36 ± 0.61	>0.05
TC (mg/dL)	182.92 ± 62.03	202.72 ± 75.48	>0.05
TG (mg/dL)	150.97 ± 78.94	162.67 ± 85.68	>0.05
Phosphate (mg/dL)	5.71 ± 1.34	5.09 ± 1.14	<0.05
PTH (pg/mL)	403 ± 250.22	272.57 ± 181.79	<0.05
Hemoglobin (g/dL)	9.89 ± 1.45	10.2 ± 1.48	>0.05
PWV (m/s)	10.17 ± 2.73	9.28 ± 1.95	>0.05
CCI (point)	6.13 ± 2.95	6.79 ± 3.28	>0.05
Causes of CKD			
	pH < 7.35 *N* = 68	pH ≥ 7.35 *N* = 48	*p* *
DM and HA (%)	33 (49)	23 (48)	>0.05
Chronic GN (%)	19 (28)	12 (25)	>0.05
ADPKD (%)	8 (12)	4 (8)	>0.05
IN (%)	3 (4)	2 (4)	>0.05
others (%)	5 (7)	7 (15)	>0.05

* χ^2^ test. Abbreviations: BMI, body mass index; pO_2_, oxygen partial pressure; pCO_2_, carbon dioxide partial pressure; HCO_3_^−^, bicarbonate; ABE, actual base excess; SBE, standard base excess; K^+^, potassium; Na^+^, sodium; Ca^2+^, ionized calcium; Cl^−^, chloride; TSH, Thyroid Stimulating Hormone; TC, total cholesterol; TG, triglycerides; PTH, parathormone; PWV, pulse wave velocity; CCI, Charlson comorbidity index; CKD, chronic kidney disease; DM, diabetes mellitus; HA, hypertension; GN, glomerulonephritis; ADPKD, autosomal dominant polycystic kidney disease; IN, interstitial nephritis.

**Table 3 diagnostics-13-03367-t003:** Clinical data obtained from a 2-year follow-up depending on the type of metabolic acidosis.

Follow-Up Data after 2 Years						
	non-MA *N* = 24	AGMA *N* = 48	HAGMA *N* = 44	F	*p*	All Study Patients *N* = 116
Time between POCT assessment and HD start (days)	161 ± 207	238 ± 229 ^a^	113 ± 184 ^b^	3.52	<0.05 *	175 ± 214
Number of patients who did not start HD (%)	2 (3)	8 (7)	3 (3)		>0.05 **	13 (11)
Number of patients who started HD (%)	21 (18)	42 (36)	39 (34)		>0.05 **	102 (87)
Number of patients who received a kidney transplant (%)	3 (3)	4 (3)	8 (7)		>0.05 **	15 (13)
Number of patients who experienced a cardiovascular event *** (%)	6 (5)	13 (11)	11 (9)		>0.05 **	30 (26)
Number of patients who died (%)	5 (4)	7 (6)	11 (9)		>0.05 **	23 (20)

Mean ± standard deviation. * ANOVA with Tukey’s post hoc test. Different letters in rows imply statistical differences at the significance level *p* < 0.05. ** Kruskal–Wallis ANOVA test, *p*-value < 0.05 was statistically significant. Abbreviations: non-MA, non-metabolic acidosis; AGMA, metabolic acidosis with a normal anion gab; HAGMA, high-anion-gab metabolic acidosis; HD, hemodialysis. *** Cardiovascular complications are listed in the Results section.

**Table 4 diagnostics-13-03367-t004:** The clinical follow-up data for the two subgroups of patients were assigned to compensated acidosis (pH > 7.35) and decompensated acidosis (pH ≥ 7.35).

Follow-Up Data after 2 Years			
	pH < 7.35 *N* = 68	pH ≥ 7.35 *N* = 48	*p*
Time between POCT assessment and HD start (days)	138 ± 149	155 ± 198	>0.05 *
Number of patients who did not start HD (%)	4 (3)	9 (8)	>0.05 **
Number of patients who started HD (%)	64 (55)	38 (33)	>0.05 **
Number of patients who received a kidney transplant (%)	8 (7)	7 (6)	>0.05 **
Number of patients who experienced a cardiovascular event *** (%)	16 (14)	14 (12)	>0.05 **
Number of patients who died (%)	14 (12)	9 (8)	>0.05 **

* Student t-test; ** χ^2^ test; *** Cardiovascular events included stroke, myocardial infarction, decompensated heart failure, atrial fibrillation, percutaneous coronary angioplasty and pacemaker implantation. Abbreviation: HD, hemodialysis.

## Data Availability

The data presented in this study are available on request from the corresponding author.

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
