# Peer review of "Point-of-Care Testing to Differentiate Various Acid–Base Disorders in Chronic Kidney Disease"

_diagnostics, 2023, doi:10.3390/diagnostics13213367_

Round 1

Reviewer 1 Report

Comments and Suggestions for Authors

The present manuscript concerns a day-to-day practical clinical subject identifying the prevalence of various acide-base disorders in patients with advanced chronic kidney disease (CKD) using point-of-care testing (POCT), and determining the relationship between POCT parameters. Interestingly, it should be noted the finding of this study that bicarbonate concentration did not correlate with renal function, which is related, as currently known, with diet. In addition, intentionally reducing protein intake can lead to an increase in tCO2. Despite the limited sample size of participating patients in advanced CKD, chloride levels did not correlate with kidney function, finding consistent with previous studies.

The language seems correct throuthout the paper and the references in general up-to-date.

However, the major limitation of this study is the lack of observed changes over time, due to its cross-sectional nature. Furthermore, important uremic toxins, such as serum sulfate derivates, have not been measured.   The authors could add 3-4 sentences concerning these remarks in order to improve describing this clinical model.

Author Response

Reviewer 1

The present manuscript concerns a day-to-day practical clinical subject identifying the prevalence of various acide-base disorders in patients with advanced chronic kidney disease (CKD) using point-of-care testing (POCT), and determining the relationship between POCT parameters. Interestingly, it should be noted the finding of this study that bicarbonate concentration did not correlate with renal function, which is related, as currently known, with diet. In addition, intentionally reducing protein intake can lead to an increase in tCO2. Despite the limited sample size of participating patients in advanced CKD, chloride levels did not correlate with kidney function, finding consistent with previous studies.

The language seems correct throuthout the paper and the references in general up-to-date.

However, the major limitation of this study is the lack of observed changes over time, due to its cross-sectional nature. Furthermore, important uremic toxins, such as serum sulfate derivates, have not been measured.   The authors could add 3-4 sentences concerning these remarks in order to improve describing this clinical model.

Response: Thank you for all remarks. The reviewer is right that major limitation of this study is the lack of observed changes over time in POCT parameters. However, as suggested by two reviewers, we made clinical follow-up and added it to the recent version of manuscript. We found that most of our patients have changed their status and are now on hemodialysis, while some have undergone transplants or died. The most significant finding was that patients diagnosed with HAGMA required renal replacement therapy to start significantly earlier than those diagnosed with AGMA.

The manuscript has been updated with appropriate tables and figures.

We also added following sentences regarding the sulphone derivates in limitation section. “We did not measure protein-bound uremic solutes because this would replicate other well-known studies on this topic. As is well-known, most patients with chronic kidney disease have metabolic acidosis and a normal serum anion gap; an increased anion gap is uncommon unless the eGFR is extremely low (i.e., less than 15 ml/min/1.73 m2) due to phosphate, sulfate, and other anion accumulation. [26] We did not test for protein-bound uremic toxins, for example indoxyl sulfate and p-cresyl sulfate, because this is not a standard parameter that is regularly assessed. Chromatography is used to measure serum total and free indoxyl sulfate as well as p-cresyl sulfate; however, this technique is not easily accessible. Our study concentrated mainly on POCT parameters and basic biochemical laboratory indices that are readily accessible to most practitioners.”

Reviewer 2 Report

Comments and Suggestions for Authors

Brief description

The authors conducted this study to identify the prevalence of various acid-base disorders in 116 patients with advanced CKD stage 4 and 5, using point-of-care testing (POCT) and to determine the relationship between POCT parameters and categorized according to the type of acidosis-base disorder. They were decompensate acidosis with pH 24 <7.35 in 68 (59%) of patients. Metabolic acidosis (MA) with HCO3- ≤22 mmol/l in 92 (79%) patients. In group of MA, AGMA and HAGMA was observed in 48 (52%) 27 and 44 (48%) of patients, respectively. Mean creatinine was significantly lower in the AGMA group compared to the HAGMA group (4.91 v 5.87 mg/dl, p<0.05). AG correlated positively with creatinine (r=0.44, p<0.01) and urea (r=0.53, p<0.01), but there was no correlation between HCO3- and both creatinine (r=-0.015, p>0.05) and urea (r=-0.07, p>0.05). Cl- concentrations correlated negatively with HCO3- (r=-0.72, p<0.05). The conclude the most common type of acid-base disturbance in CKD patients stages 4 and 5 is AGMA, which is observed in patients with better kidney function and is associated with compensatory hyperchloremia. The more advanced the CKD, the higher the AG.

Comments.

1.      The aim of study is obscure in this manuscript. Most of the findings had been well described in the literatures and textbooks, so it does not increase our understanding of the metabolic acidosis in CKD patients, and also no new finding that is valuable to our knowledge.

2.      In this type of study, it is very important to know the accuracy of the laboratory machine of the POCT. Unfortunately, we just can find the machine name, but there is no other information about how well it works.

3.      The blood was drawn from artery while patients received surgical creation of arterio-venous fistula for preparation of dialysis at CKD stage 4 or 5. Though how early the shunt should be created is still a controversial issue and not the main focus of study, it seems reasonable to see the relationship between HCO3 vs Cl-, and AG vs Cr (UN or eGFR), but there is nothing new to us.

4.      From these findings and comment, I would suggest that the authors should revise their MS and add some clinical important events in relation to degree of metabolic acidosis, which will increase it value in management of advance CKD with metabolic acidosis problem.

Comments on the Quality of English Language

NO

Author Response

Reviewer 2

Brief description

The authors conducted this study to identify the prevalence of various acid-base disorders in 116 patients with advanced CKD stage 4 and 5, using point-of-care testing (POCT) and to determine the relationship between POCT parameters and categorized according to the type of acidosis-base disorder. They were decompensate acidosis with pH 24 <7.35 in 68 (59%) of patients. Metabolic acidosis (MA) with HCO3- ≤22 mmol/l in 92 (79%) patients. In group of MA, AGMA and HAGMA was observed in 48 (52%) 27 and 44 (48%) of patients, respectively. Mean creatinine was significantly lower in the AGMA group compared to the HAGMA group (4.91 v 5.87 mg/dl, p<0.05). AG correlated positively with creatinine (r=0.44, p<0.01) and urea (r=0.53, p<0.01), but there was no correlation between HCO3- and both creatinine (r=-0.015, p>0.05) and urea (r=-0.07, p>0.05). Cl- concentrations correlated negatively with HCO3- (r=-0.72, p<0.05). The conclude the most common type of acid-base disturbance in CKD patients stages 4 and 5 is AGMA, which is observed in patients with better kidney function and is associated with compensatory hyperchloremia. The more advanced the CKD, the higher the AG.

Comments.

  1. The aim of study is obscure in this manuscript. Most of the findings had been well described in the literatures and textbooks, so it does not increase our understanding of the metabolic acidosis in CKD patients, and also no new finding that is valuable to our knowledge.

Response:  Thank you for your valuable feedback on our manuscript. We appreciate your concern about the clarity of our aim statement and the novelty of our findings. In the introduction, we additionally described the traditional causes of high anion gap metabolic acidosis according to ref. [1] (Fenves, A.Z.; Emmett, M. Approach to Patients With High Anion Gap Metabolic Acidosis: Core Curriculum 2021. Am J Kidney Dis 2021, 78, 590-600, doi:10.1053/j.ajkd.2021.02.341.) and noted that general principles regarding metabolic acidosis may not apply to advanced CKD. Our study's primary objective was to reevaluate this conventional viewpoint.

The reviewer is right in stating that the results presented in the primary version of the manuscript are similar to those in other papers. On the other hand, in this study we used a particular technique to obtain blood for POCT testing, and the study group was unique because of its homogeneity. Additionally, as suggested by two reviewers, we performed a 2-year follow-up of clinical data, analyzed the results, and found interesting new findings. Based on this, we revised our aims, and rephrasing it to make it more explicit. In addition, we highlighted the significance of our results by outlining the implications for clinical practice and further investigation. We hope that these changes address your concerns and improve the quality of our manuscript.

Three aspects in the revised text set the manuscript apart from others:

  • In the study, we used a new method of collecting arterial blood for POCT directly from the artery during the creation of an arteriovenous fistula. Arterial blood provides more information about acid-base disorders than venous. Our method avoids additional puncture of the artery, which reduces the risk of complications, e.g. hematomas or arterial pseudoaneurysms. To our knowledge, this is the first study describing this method of arterial blood collection for POCT.
  • Our study includes patients in the most advanced stages of CKD (Stages 4 and 5) with the most severe acid-base disorders that are not completely controlled. Insufficient information from the attending physician regarding dietary restrictions and bicarbonate supplementation, as well as patients' failure to adhere to the diet, are contributing factors. 79% of our patients have metabolic acidosis, predominantly AGMA (48%), but there are also patients with HAGMA (44%), who require a more aggressive diet and bicarbonate supplementation. This study demonstrated that metabolic acidosis is a neglected area of nephrology that requires improvement. We included this information in Discussion section.
  • Based on the reviewer's suggestions, we conducted a 2-year follow-up. It was found that patients assigned to the HAGMA group started renal replacement therapy considerably earlier than those in the AGMA group. The practical significance of this conclusion stems from the fact that if a patient is classified as HAGMA, the nephrologist is obligated to speed up making plans for renal replacement therapy, including the creation of a vascular access. Naturally, this finding should be interpreted with caution because the qualification for starting dialysis is multifactorial and the decision is influenced by, clinical features of overhydration, high nitrogen metabolites values, hyperpotassemia, etc.

2. In this type of study, it is very important to know the accuracy of the laboratory machine of the POCT. Unfortunately, we just can find the machine name, but there is no other information about how well it works.

Response: Your remark is very important. The RADIOMETER ABL90 SERIES, RADIOMETER MEDICAL APS, Denmark POCT devices is fully automatic. It has a cartridge containing test reagents and performs an 8-hour calibration. This information was added to the text.

  1. The blood was drawn from artery while patients received surgical creation of arterio-venous fistula for preparation of dialysis at CKD stage 4 or 5. Though how early the shunt should be created is still a controversial issue and not the main focus of study, it seems reasonable to see the relationship between HCO3 vs Cl-, and AG vs Cr (UN or eGFR), but there is nothing new to us.

Response: We agree that the timing of shunt placement was not the primary focus of this study and in fact depends on GFR decline or clinical indications (e.g., diuretic-resistant fluid accumulation). In the Clinic, we try to create ESKD Life-Plan and prepare vascular access when eGFR is 15-20 mL/min/1.73 m2 in accordance with current guidelines (https://www.ajkd.org/article/S0272-6386(19)31137- 0/fulltext). One of the important elements of the assessment of a patient with CKD is the assessment of acid-base disorders. In the study we utilized the possibility of collecting arterial blood during AVF creation because it identify such disorders more precisely than venous blood. To our knowledge, this type of diagnostics has never been used for POCT collection. It has the advantage that it is not associated with arterial trauma, e.g. hematoma or other complications, but it provides important information regarding acid-base disorders, which are extremely important for patients with CKD.

We appreciate the Reviewer's suggestion to conduct a follow-up. On this basis, we concluded that in patients with HAGMA, renal replacement treatment should be initiated earlier than in patients with AGMA. Of course, there are many restrictions to this result, which should be the topic of further extensive investigation, but it demonstrates a general trend that the profound disorders associated with acidosis, the sooner renal replacement treatment should be initiated.

  1. From these findings and comment, I would suggest that the authors should revise their MS and add some clinical important events in relation to degree of metabolic acidosis, which will increase it value in management of advance CKD with metabolic acidosis problem.

Response: In accordance with the Reviewer's recommendation, an analysis was conducted on all comorbidities based on the type of metabolic acidosis and included in Table 1.

Additionally, we conducted a 2-year follow-up and checked the clinical status of each patient based on medical records and telephone information. Follow-up time ranged from 15-51 months (mean 31 months). Within this time, we recorded: time of dialysis initiation, cardiovascular events, deaths, and kidney transplantation. The time interval between fistula creation, i.e. taking POCT collection, and starting dialysis treatment was 3-913 days, with an average of 187 days. 13 patients did not start renal replacement therapy, including 4 who were lost to follow-up. During the follow-up period, 102 patients started dialysis treatment. There were 23 deaths among the patients. It was found to be associated with cardiovascular events in 12 cases, infectious complications in 8 cases, and malignancy in 3 cases. Kidney transplantation was performed in 15 patients. Cardiovascular complications occurred in 30 patients and included: myocardial infarction, need for coronary angioplasty or coronary revascularization, paroxysmal atrial fibrillation, heart failure decompensation, ischemic and hemorrhagic stroke. Table 3 shows the number of individual complications depending on the type of metabolic acidosis. We found that patient with HAGMA have significantly shorter time to start renal replacement therapy in comparison to AGMA group (Figure 2).

The text has been changed in accordance with this information.

Reviewer 3 Report

Comments and Suggestions for Authors

The authors aimed to identify the prevalence of various acid-base disorders in patients with advanced CKD using point-of-care testing (POCT) and to determine the relationship between POCT parameters.

The abstract provides a good overview of the study's objectives and some key findings.

Introduction: Well written. Informative.

Methods:

-         The inclusion and exclusion criteria are well-defined. However, It would be helpful to mention why patients on hemodialysis were excluded.

-        The description of the nutrition/malnutrition status assessment is somewhat brief.

-        You might want to provide more details about the Nutritional Risk Score 2002 (NRS) and Subjective Global Assessment (SGA) tools used and how these assessments were conducted.

-        Explain why the four patients receiving bicarbonate were instructed to discontinue its use three days before surgery.

-        Provide a bit more detail about the point-of-care testing (POCT) procedure, especially how the arterial blood sample was obtained, and any specific protocols followed during the procedure.

Results:

I do appreciate the efforts in presenting the results of the study. I just recommend that you mention the only significant results of the post-hoc analysis, without the need for the table. The figures are informative.

Discussion:

Robust, well written and to the point.

Conclusion: I suggest instead of phrases like "we showed that," you use more precise language, such as "Our study demonstrated that..." This makes your statements more authoritative. And I suggest you end the passage with a concise summary of your key findings. This helps reinforce the main takeaways for the reader. “These observations shed light on the evolving nature of acid-base disturbances in CKD, highlighting the importance of monitoring both anion gap and chloride levels in advanced CKD patients for a comprehensive understanding of their metabolic acid-base status.”

Author Response

Reviewer 3

The authors aimed to identify the prevalence of various acid-base disorders in patients with advanced CKD using point-of-care testing (POCT) and to determine the relationship between POCT parameters.

The abstract provides a good overview of the study's objectives and some key findings.

Introduction: Well written. Informative.

Methods:

-         The inclusion and exclusion criteria are well-defined. However, It would be helpful to mention why patients on hemodialysis were excluded.

Response: Thank you for your insightful comments on our manuscript. To create a more homogeneous cohort, we decided to remove hemodialysis patients. Bicarbonate concentrations in dialysis fluid are high (typically 32 mmol/l), resulting in quick correction of metabolic acidosis during hemodialysis. It would be difficult to interpret the results in a heterogeneous study group (HD patients + non HD patients).

-        The description of the nutrition/malnutrition status assessment is somewhat brief.

-        You might want to provide more details about the Nutritional Risk Score 2002 (NRS) and Subjective Global Assessment (SGA) tools used and how these assessments were conducted.

Response: The Nutritional Risk Score 2002 (NRS) is a short screening tool that assesses the risk of malnutrition in hospitalized patients. It consists of two parts: a nutritional screening section and a nutritional assessment section. The screening section includes questions about weight loss, body mass index, and food intake. The assessment section includes questions about disease severity, age, and dietary intake. Regarding the Subjective Global Assessment it is a clinical tool that assesses the nutritional status of patients based on their medical history, physical examination, and laboratory data. It is used to identify patients who are at risk of malnutrition and was validated in CKD 4-5 patients. Both tools were used at admission to hospital and none of patients showed malnutrition.

-        Explain why the four patients receiving bicarbonate were instructed to discontinue its use three days before surgery.

Response: In a few patients we decided to discontinue bicarbonate before surgery just to conduct a study on a homogeneous group of patients (uncorrected acidosis). This protocol was approved prior to the start of the study.

-        Provide a bit more detail about the point-of-care testing (POCT) procedure, especially how the arterial blood sample was obtained, and any specific protocols followed during the procedure.

Response: Arterial blood samples were collected during the creation of the AVF. After dissecting the radial artery (RA) and clamping the distal and proximal sections, a longitudinal incision was made. Blood was then drawn into a heparinized syringe after the clamp was released using a plastic needle inserted into the proximal part of the RA. The time between RA closure and sample collection was no longer than 30 seconds. The assisting nurse then took the sample to the POCT in an adjacent room and immediately performed the test. The total time from sample collection to measurement did not exceed 3 minutes. This information was added to the Methods section.

Results:

I do appreciate the efforts in presenting the results of the study. I just recommend that you mention the only significant results of the post-hoc analysis, without the need for the table. The figures are informative.

Response: Thank you for your suggestion. The table that presented a summary of the post hoc analysis results has been removed, and the relevant information was added to the text.

Discussion:

Robust, well written and to the point.

Response: Thank you for your opinion.

Conclusion: I suggest instead of phrases like "we showed that," you use more precise language, such as "Our study demonstrated that..." This makes your statements more authoritative. And I suggest you end the passage with a concise summary of your key findings. This helps reinforce the main takeaways for the reader. “These observations shed light on the evolving nature of acid-base disturbances in CKD, highlighting the importance of monitoring both anion gap and chloride levels in advanced CKD patients for a comprehensive understanding of their metabolic acid-base status.”

Response: Your suggestions are indeed helpful and improve quality of paper.